# One-Year Functional Evaluation of Patient After CD34+ Stem Cell Treatment for Hip Osteoarthritis

**DOI:** 10.3390/jpm14121126

**Published:** 2024-11-28

**Authors:** Eleonora Stefańska-Szachoń, Kamil Koszela, Marta Woldańska-Okońska

**Affiliations:** 1Faculty of Medical and Health Sciences, Radom University, 26-600 Radom, Poland; e.stefanska-szachon@uthrad.pl; 2Department of Neuroorthopedics and Neurology Clinic and Polyclinic, National Institute of Geriatrics, Rheumatology and Rehabilitation, 02-637 Warsaw, Poland; 3Department of Internal Medicine, Rehabilitation and Physical Medicine, Medical University of Lodz, 90-419 Lodz, Poland; marta.okonska@poczta.onet.pl

**Keywords:** hip osteoarthritis, CD34+ stem cells, physiotherapy, Harris questionnaire

## Abstract

**Background/Objectives**: The etiology and causes of osteoarthritis are still being studied at the cellular and molecular level by many scientists around the world. With advances in knowledge, technology, and the need for complexity, new therapeutic approaches—such as restorative medicine—are being developed to protect the patient from endoprosthesis implantation, aiming to simultaneously restore and maintain physical and psychosocial function. The purpose of this study was to evaluate the effectiveness of physiotherapy after the implantation of CD34+ stem cells into the hip joints of patients with osteoarthrosis. **Methods**: The study included 71 patients, previously diagnosed with HOA (hip osteoarthritis) and undergoing CD34+ stem cell therapy followed by rehabilitation. The twelve-month prospective follow-up included 23 women and 48 men. Participants were asked to provide information, regarding the quality of their daily functioning, according to the Harris Questionnaire at four time points. **Results**: In all evaluations and groups of patients, i.e., women and men, and rehabilitated and non-rehabilitated, a significantly high improvement was noted at 3 months after the administration of CD34+ stem cells, while afterwards there was a slight decrease in the value of the results obtained at 6 months after the procedure, which improved at further stages of the study. The value of functioning on the Harris scale did not differ significantly between genders, although there was a trend of better functioning in men after one year, while it was higher in rehabilitation subjects. **Conclusions**: Patients’ daily functioning was better in the rehabilitation group, as reflected by the Harris scale. The absence of adverse symptoms and the small differences in physiotherapy results between the rehabilitated and non-rehabilitated groups testify to the high effectiveness of the stem cell therapy method.

## 1. Introduction

Continuous advances in medicine are focused on the most common diseases, especially those of the human osteoarticular system. In the 21st century, many diseases, classified as “civilization” diseases, are still a serious problem, negatively affecting the quality of life of patients. Ailments from the osteoarticular system account for a significant percentage of them, and painful symptoms are observed among an increasingly younger population.

Considering the prevalence of osteoarthritis (OA), unfavorable data from research findings, clinical observations, and statistical results in recent years prove that OA is one of the most common ailments and is a major cause of disability [1].

Despite ongoing research, the specific pathophysiology of the development of CAD cannot be completely determined with exactitude. It is a multifactorial, dynamically progressive disease process that leads to significant disparities in the formation and degeneration of both joint and periarticular structures. Although the characteristic pathological feature of OA is the loss of articular cartilage, OA is widely recognized as a disease involving the whole joint, including ligaments, meniscus, synovitis, and the joint capsule [2]. The disease has the hallmarks of the chronic inflammation of the whole joint, which leads to progressive softening, fibrosis, ulceration, and the loss of articular cartilage, as well as sclerosis and the thickening of the subchondral bone tissue, with the formation of numerous cysts and pathophysiological bone growths in the form of osteophytes.

All these changes eventually lead to the reduced mobility of the joint, projecting increasing soreness and impairing the ability to perform basic activities of daily living. In addition, they reduce the qualities of social participation, having a destructive effect on a person’s overall psychological state [2,3].

The localization of OA in the joints of the lower extremities, particularly the hips, accounts for a high percentage of disability, with a noticeable predominance among female patients and with high body weight, leading to obesity, observed in both sexes [3,4]. The high percentage of OA incidence among women is explained by hormonal changes occurring in the female body during menopause, which is also relevant in observing fluctuations in body mass indexes, which can imply slowed metabolism and disrupted metabolism [5,6,7,8,9].

Recent studies show that OA is not an ailment that only affects the elderly, as the age of the joint, specifically the cartilage, does not cause the disorder. The inflammatory process of OA begins in the synovial membrane with the activation of the immune system, involving both humoral and cellular mediators. The so-called “damage-associated molecular patterns” (DAMPs) play a key role in this process [1]. Of great importance in etiology are the genetic background and microbial characteristics, such as mutations in the genes encoding cartilage matrix elements, signal transduction mediators, and the cytokines responsible for maintaining articular cartilage homeostasis, as well as mutations in the amino acid sequences of collagen polypeptide chains and chondrocyte apoptosis [10,11,12,13,14]. Circulatory changes involving decreased venous outflow in subchondral bone also contribute to the progression of OA, resulting in physicochemical changes that stimulate osteoblasts to express bone remodeling and cartilage-damaging cytokines [1]. TGF-β and IL-6 are SASP (senescence-associated secretory phenotype) factors that can contribute to chondrocyte aging by activating p15, p21, and p27 proteins, thereby enhancing aging through the SMAD complex or STAT3 pathway [1]. In addition, SASP release by aging chondrocytes exerts a chemotactic effect on immune cells, creating an inflammatory environment that further stimulates cartilage degradation [1].

It is estimated that some form of OA occurs in 25% of the general population. In the Polish population, the rate is 15–20%. Statistical scientific sources from the United States report that the rate of people with osteoarthritis of the hip (OA) around the age of 40–45 is already almost 10% of the population. As these data show, OA is affecting younger and younger people and is one of the most prominent civilization diseases, on par with cardiovascular and metabolic ailments. As a result, it is a major challenge for modern medicine and financial policy, especially in Poland, which is explained by rising inflation and medical costs [15].

The long waiting time for specialized treatment causes a buildup of health problems and leads to negative effects on the psyche, so any disability acquired will take on the dimensions of a global psycho-physical and sociological–social problem [16].

The risk factors of HOA, osteoarthritis of the hip, are manifold and can exist both individually and together, and their possible interrelationships are defined in Figure 1 [1,2,17,18].

The first symptom of osteoarthritis is the sensation of pain. In BD, it can be of varying degrees of severity, appearing in a selective, local manner or being felt in the whole joint. In the advanced form of CDSB, it prevents nightly rest and leads to permanent, multisite somatic changes, caused by sedentary lifestyle and taking increasingly powerful drugs to minimize pain [19]. The progressive disease process successively reduces overall physical fitness.

The diagnosis of coxarthrosis should include a physical and clinical examination, including taking a detailed history [20,21,22]. The Harris Hip Score (HHS) scale is the most commonly used to assess a patient’s functional hip joint [23].

Every few years, international scientific societies develop and publish the latest recommendations for the treatment of OA. The subject of attention for some time has been the personalization of treatment, directed toward an individualized approach to the patient. For a few years now, stem cell treatment has been an innovative therapeutic method for HAO; however, it is still under discussion. The method aims to postpone the need for an endoprosthesis by regenerating joint cartilage and minimizing pain sensations. Mesenchymal stem cells (MSCs) may be a promising option among all other therapeutic options. However, many issues are still debated, such as the best source of the cells, their nature, and the right amount [1]. Stem cells (SCs) have the ability and capacity to self-differentiate (plasticity), transform into any cell of the body, and show specific potential for proliferation (self-renewal). In the adult body, we can find mesenchymal stem cells (MSCs), which have proliferative properties, can be differentiated into a specific cell type of the whole organism, and can be obtained from various sources, such as adipose tissue, bone marrow, umbilical cord blood, and peripheral blood. Hematopoietic stem cells (HSCs) are found in the bone marrow and give rise to all blood cells [24,25].

One of the first stem cell researchers in the world was James Alexander Thomson. Further large-scale efforts in this field were made, among others, by Irving Weissman at Stanford University [26], who, together with his team of researchers, selected many populations of human progenitor cells, including the CD34+ lineage. The CD34 antigen is described as an early marker of bone marrow renewal and is also detectable in the liver, spleen, and thymus [27].

The therapy process involves the simultaneous collection of the patient’s peripheral blood and separation of CD34 stem cells. The next stage is carried out in the operating theater, where anesthesia is administered to the spinal canal and the previously collected preparation of selected CD34+ cells is injected into the joint under ultrasound guidance. This is followed by an immediate rehabilitation process; that is, bringing the limb to physiological ranges of mobility, which is made possible by the patient’s lack of pain and mobility restrictions. Further therapeutic management begins the day after the procedure.

The aim of this study was to evaluate the effectiveness of physiotherapy after implantation of CD34+ stem cells into the hip joints.

## 2. Materials and Methods

The study included 71 patients, previously diagnosed with hip osteoarthritis (Kellgren–Lawrence scale grade II) and undergoing CD34+ stem cell therapy. The 12-month prospective follow-up included 23 women and 48 men (Figure 2) who were qualified for CD34+ stem cell therapy and gave written consent to participate in the study.

The mean age of the included patients was 57.43 years (SD = 12.70 years) and, despite apparent differentials by gender, was not statistically significantly different in the range in question (*p* = 0.2240). The mean value of body mass index (BMI) was 28.68 (SD = 4.45) kg·m^−2^, also without statistically significant difference by sex of the patients studied (*p* = 0.1693). The detailed descriptive measures for age and BMI are shown in Table 1. Patients who did not give written consent were not eligible for the study.

Each participant in the project underwent the identical treatment procedure of implanting stem cells into the hip joint and starting rehabilitation. In further stages, physical therapy was continued. Rehabilitation consisted of three parts, as follows: 1. Hospital rehabilitation (7 days) based on the use of a CPM (Continuous Passive Motion) splint, non-weight-bearing exercises, sollux lamp (blue light). In total, approx. 1.5 h per day. 2. Outpatient rehabilitation (6 weeks) based on the use of exercises improving hip joint mobility, active exercises with resistance, and quadriceps electrotherapy. In total, approx. 1.5 h per day, 5 times per week. 3. Home rehabilitation (rest of the time), involving the continuation of learned exercises. In total, approx. half an hour per day, 5 times per week. This course of therapy was for hip osteoarthritis patients qualified for CD34+ stem cell therapy, from the time the cells were harvested and implanted into the joint until the end of the 12-month period, after which the procedure was evaluated. The patients’ quality of life was assessed, for example, by participation in the activities of daily living at defined time intervals: just after cell implantation, and after 3, 6, 9, and 12 months. From the presented study population, a comparison group (the so-called non-rehabilitated group), consisting of 12 subjects (4 women and 8 men), was selected, which did not report a desire to continue systematic rehabilitation, but agreed to systematic follow-up examinations at the mentioned time intervals. Participants were asked to provide updated information regarding their quality of daily functioning by answering questions according to the Harris Questionnaire [23].

The Harris Hip Score (HHS) scale consists of several parts, each of which determines different values. The first includes questions about the pain the patient experiences on a daily basis; in the second, the patient determines the distance he or she is able to walk without resting. The next questions concern self-care directed at putting on socks and shoes, the ability to get around using public transportation, the need for orthopedic aids and supplies, observing any limp when walking, climbing stairs, and how long it takes to assume a pain-free sitting position in a low chair and a high chair [23].

In the next part of the questionnaire, there is an opportunity to answer “yes” or “no” to questions about functional status. If the respondent meets all four criteria listed, the therapist marks a “yes” answer; if even one of the criteria is not met, then a negative answer should be marked. The last part of the Harris Questionnaire is mobility, to accurately determine the range of motion in the hip joint [23]. The patient can score a maximum of 100 points on the entire scale. Functional status is determined according to the established scoring standards:

-Very good score—91–100 points,-Good score—81–90 points,-Moderate score—71–80 points,-Insufficient score—70 points and below.

### Statistical Analyses

Measurable (numerical) characteristics were described using measures of central tendency and dispersion, with 95% confidence intervals and minimum and maximum values of the variables. Shapiro–Wilk’s W test and Levene’s test were initially applied. A multivariate analysis of variance (ANOVA) with repetition was performed for variables with a normal distribution. Generalized linear models with repeated measures were used for traits with a non-normal distribution. *p* < 0.05 level was considered statistically significant.

## 3. Results

The functioning of HOA patients, as assessed by the Harris scoring scale, changed in a statistically significant manner in both female (*p* < 0.0001) and male (*p* < 0.0001) subjects.

-Women—on average from 76.23 points (SD = 16.41 points) to 80.22 points (SD = 18.65 points), representing a relative improvement of 5.23%.-Men—on average from 78.50 points (SD = 19.57 points) to 88.13 points (SD = 13.90 points), representing a relative improvement of 12.27%.

The dynamics of change in the functioning of patients with HOA was not statistically significantly different according to the gender of the patients studied (*p* = 0.2713), (Figure 3, Table 2), although there is a trend of better functioning in men after one year.

## 4. Discussion

During the observation of the evaluation of the physiotherapy results of HAO patients according to the Harris scale and the amplitude of the variables in the percentage conversion of the defined score, the values in both sexes increased steadily over 1 year, with a lower increase in women (+5.23%) than in men (+12.27%). It is noteworthy that in men, the increase in values occurred consecutively during each study period until the end of the medical experiment, while in women there was a decrease after 6 months (Figure 3, Table 2).

Similarly, in both the R and NR comparison groups, the significant improvement in score values culminated at 3 months after the administration of CD34+ stem cells, with a dramatic increase at this time in the NR group. The R group showed an improvement in values noted up to the 6th month of the trial, and then a slight decrease in values was observed at the 12th month. In NR subjects, a decrease in values was already noted at both time follow-ups, i.e., at 6 months and 1 year after treatment (Figure 4, Table 3).

The main element highlighted in the cited studies is the efficacy and safety in treating early moderate osteoarthritis of the hip in the long term, with improvements in the clinical condition of the study participants and overall pain reduction. In addition, no adverse effects were observed in patients during the follow-up period, demonstrating the safety profile of this procedure [28].

Indeed, the natural course of action should be a process of systematic physical participation, which will certainly translate into improved quality of function. This is shown by the results presented in this paper in both R and NR patients. The observed overall decrease in the quality of functioning in women, at the last stage of the experiment, can be explained by many individual considerations. The unequivocal claim that there will be a deterioration in daily functioning in the subsequent years after hip regeneration with CD34+ stem cells is not true. A case report of a 45-year-old woman diagnosed, in infancy, with congenital hip dislocation, published in Physiotherapy Poland, can be cited here. Many methods of modern physiotherapy have been of limited help, and undergoing treatment with CD34+ stem cells resulted in an improvement in the patient’s quality of life and allowed her to keep her own hip joint for a decade [28,29].

Attempts to use well-known, albeit not always effective, imaging-based treatments (X-ray, MRI) are detrimental to the often well-functioning individual, in whom too little attention is paid to clinical status and functional tests. Diagnostic imaging still remains an additional test, sometimes systematizing the results obtained or facilitating diagnosis. Nevertheless, it should not be solely an indication for surgery. More and more researchers, especially in the United States of America, but also in Europe (Italy, Poland), are focusing attention on the still admittedly short-lived but positive results of treatment of osteoarthritis of the hip with stem cells [30,31,32,33]. The direction of osteoarthrosis research in modern medicine is focused on restorative medicine, in which, in addition to the use of stem cells, hyaluronic acid [34] or platelet-rich plasma is simultaneously introduced into the joint. Functional and clinical evaluations report favorable results 5 years after surgery. This may push back the need for endoprosthesis into the indefinite future. Trials of exosome therapy are also being carried out. A substance called Wharton’s Jelly (Wj), derived from the umbilical cord (UC), has gained interest from researchers.

Although Wj-derived mesenchymal stem cells (MSCs) are tentatively promising for joint cartilage restoration, they also have some limitations. Wj-derived exosomes, on the other hand, have a strong ability to participate in reducing synovial inflammation and the progressive destruction of the osteocyte environment. Identifying the content of exosomes in future studies may help determine the anti-inflammatory mechanisms of exosomes in various diseases and be applied to therapeutic approaches [35].

A limitation of the study is the lack of a control group without CD34+ stem cell treatment for hip osteoarthritis. The present study reports only functional values. It would be worthwhile in the future to analyze these results in correlation with imaging studies, in order to establish clearer indications for the procedure of administering stem cells with a scaffold to joints affected by osteoarthrosis.

## 5. Conclusions

The physiotherapy treatment regimen used appears to be an effective way to improve patients’ functioning after CD34+ stem cell implantation.

The daily functioning of the patients was better in the rehabilitation group, as reflected by the trend in the Harris scale.

The persistence of better functional outcomes between the sexes after one year is probably due to differences in the physiology of aging, which may result in the rehabilitated men being younger than the women, had better outcomes at the beginning of the research experiment, and were probably related to less sarcopenia.

The absence of adverse symptoms and the small differences in physiotherapy results between the rehabilitated and non-rehabilitated groups may attest to the effectiveness of the stem cell therapy method.

## Figures and Tables

**Figure 1 jpm-14-01126-f001:**
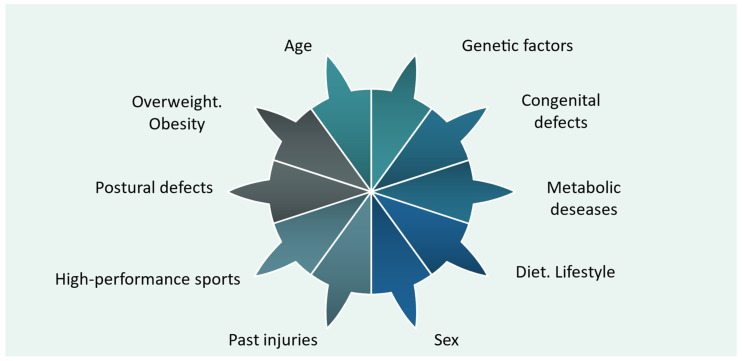
Risk factors for osteoarthritis [1,2,17,18].

**Figure 2 jpm-14-01126-f002:**
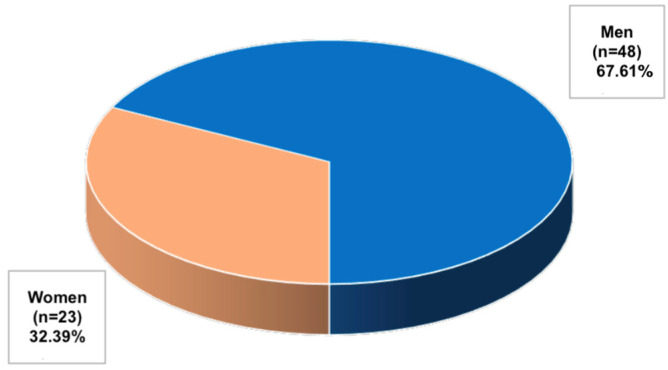
Structure of the surveyed patients by gender.

**Figure 3 jpm-14-01126-f003:**
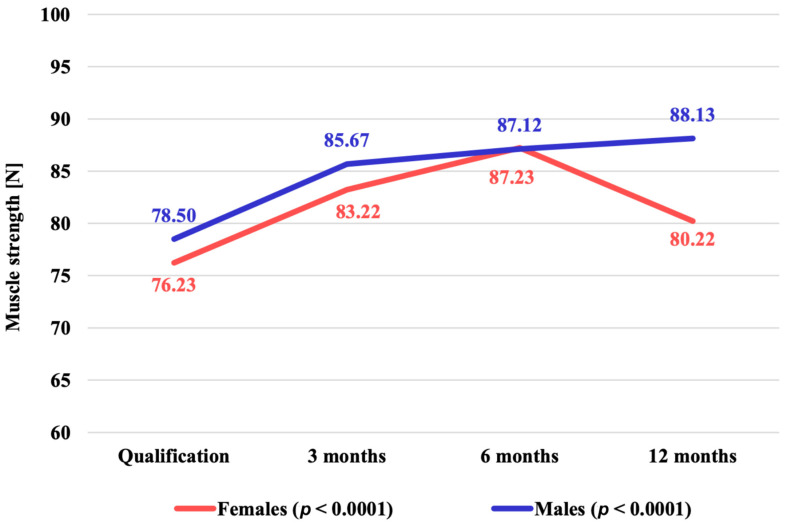
Mean values of the Harris scale functional assessment of HOA patients (points) over 12 months of follow-up in the studied patients overall and by gender (*p* = 0.2713).

**Figure 4 jpm-14-01126-f004:**
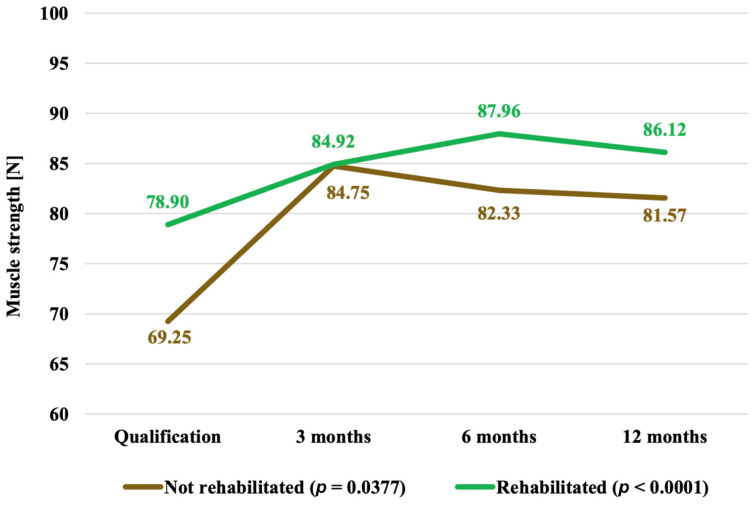
Mean values of the Harris scale functional assessment of HOA patients (points) over 12 months of follow-up in non-rehabilitated versus rehabilitated patients studied (*p* = 0.4728).

**Table 1 jpm-14-01126-t001:** Descriptive statistics of age (years) and BMI of study patients overall and by gender (n = 71).

	Sex	Statistical Parameters						
		M	SD	Me	Q_1_–Q_3_	95% CI	Min.–Max.	*p*
Age (in years)	Women	61.33	9.47	62.00	54.00–68.00	55.32–67.35	44.00–76.00	
	Men	55.39	13.85	58.00	48.00–64.00	49.40–61.38	25.00–77.00	=0.2240
	Total	57.43	12.70	60.00	48.00–66.00	53.06–61.79	25.00–77.00	
BMI (kg·m^−2^)	Women	27.42	4.12	27.50	25.25–29.00	24.80–30.04	19.00–36.00	
	Men	29.28	4.55	2.00	25.50–29.00	27.40–31.16	22.00–38.00	=0.1693

Definition of the statistical symbols used: M—mean-average; SD—standard deviation; Me—median; Q—quartile; CI—confidence interval; *p*—*p*-value—probability value, or level of statistical significance.

**Table 2 jpm-14-01126-t002:** Descriptive statistics of the functional assessment of HOA patients according to the Harris scale (scores) over 12 months of follow-up in total patients/and by gender (n = 71).

Time of Examination	Physical Therapy	Statistical Parameters						
		M	SD	Me	Q_1_–Q_3_	95% CI	Min.–Max.	*p **
Before treatment	No	69.25	25.25	67.50	48.50–93.50	48.14–9.36	36–99	
	Yes	78.90	17.39	82.00	67.00–93.50	74.41–83.39	24–99	
After 3 months	No	84.75	20.17	90.50	82.50–97.00	67.89–10.00	38–100	
	Yes	84.92	14.81	92.00	76.00–95.00	80.66–89.17	45–100	
After 6 months	No	82.33	21.16	91.00	73.00–97.00	66.07–98.60	39–100	
	Yes	87.96	13.00	93.00	86.00–96.00	84.41–91.51	47–100	
After 12 months	No	81.57	19.10	92.00	72.00–95.00	63.90–99.24	46–100	=0.0377
	Yes	86.12	15.51	92.00	88.00–95.50	81.62–90.63	46–100	<0.0001

Definition of the statistical symbols used: M—mean–average; SD—standard deviation; Me—median; Q—quartile; CI—confidence interval; *p*—*p*-value—probability value, or level of statistical significance; * multivariate analysis with repeated measures, controlled for age (*p* = 0.0032), gender (*p* = 0.2713) and BMI (*p* = 0.1869) of study patients.

**Table 3 jpm-14-01126-t003:** Descriptive statistics of the Harris scale assessment of HOA patients’ functioning (scores) over 12 months of follow-up in non-rehabilitated versus rehabilitated patients studied (n = 71).

Time of Examination	Physical Therapy	Statistical Parameters						
		M	SD	Me	Q_1_–Q_3_	95% CI	Min.–Max.	*p **
Before treatment	Women	76.23	16.41	79.50	6300–90.00	68.95–83.50	44–98	
	Men	78.50	19.57	83.50	67.00–95.00	72.69–84.31	24–99	
	Total	77.76	18.51	81.00	65.00–93.50	73.28–82.24	24–99	
After 3 months	Women	83.22	15.72	89.00	60.00–94.00	75.40–91.04	45–100	
	Men	85.67	15.48	92.00	78.00–96.00	80.65–90.68	38–100	
	Total	84.89	15.46	92.00	78.00–95.00	80.79–89.00	38–100	
After 6 months	Women	87.23	14.63	9350	86.00–96.00	80.74–93.72	47–100	
	Men	87.12	14.40	92.00	86.00–9.00	82.58–91.67	39–100	
	Total	87.16	14.36	93.00	8600–96.00	83.54–90.78	39–100	
After 12 months	Women	80.22	18.65	88.50	6200–94.00	70.95–89.49	46–100	<0.0001
	Men	88.13	13.90	93.00	88.00–95.00	83.50–92.77	46–100	<0.0001
	Total	85.55	15.89	92.00	84.0.0–95.00	81.25–89.84	46–100	<0.0001

Definition of the statistical symbols used: M—mean–average; SD—standard deviation; Me—median; Q—quartile; CI—confidence interval; *p*—*p*-value—probability value, or level of statistical significance. * multivariate analysis with repeated measures, controlled for age (*p* = 0.0032), gender (*p* = 0.2713) and BMI (*p* = 0.1869) of study patients.

## Data Availability

The data are available from the corresponding author if required.

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
