# Peer review of "One-Year Functional Evaluation of Patient After CD34+ Stem Cell Treatment for Hip Osteoarthritis"

_jpm, 2024, doi:10.3390/jpm14121126_

Round 1

Reviewer 1 Report

Comments and Suggestions for Authors

The authors report their own experience of intra articular injection of CD34+ stem cells in patients suffering from hip OA. 71 patients have been included and a follow up at 12 months

Though there number of limitation in this kind of retrospectice study, it is still interesting to record the huge experience of the authors, particularly for hip OA.

General comments

The improvement in the Harris Hip Score ( HSS)  start at month  3 and is more and less stable overtime time up to month 12 , which is a little bit surprising . Moreover, the magnitude of improvement is rather limited. And finally what is the part of rehabilitation in this improvement ? 

 Specific comments

 1. There is no data on  X ray  stage hip OA ? as well as on duration of the disease , previous treatments,  others localisations of OA  etc .. it is a little short .... 

2. How many patients with unilateral versus bilateral hip OA

3. What's mean rehabilitaion process ? it consists in which kind of t program in terms of time, type of exercices and frequency ?

4. Was intake of others painkillers= , NSAIDS  recorded?

5. What do we know excatly about the HSS sensitivity to change ?

 Please comment on those different points.

Author Response

Dear Reviewer,

thank you very much for your time and all your valuable comments.

We have made the necessary changes to the manuscript according to your suggestions.

General comments
The improvement in the Harris Hip Score ( HSS)  start at month  3 and is more and less stable overtime time up to month 12 , which is a little bit surprising . Moreover, the magnitude of improvement is rather limited. And finally what is the part of rehabilitation in this improvement ? 

This is a valid comment by the reviewer. The question cannot be answered unequivocally, due to the fact that after 3 months there may have been different environmental factors, although patients were given the same management recommendations. 

Moreover, the process of intensive rehabilitation in the hospital (1 week) and outpatient rebailitation (6 weeks) lasted 7 weeks in total. Secondly, the effects of rehabilitation are visible a few weeks after its completion. During institutional rehabilitation, conditions are the same for all patients. It is more intensive.

On the other hand, improvement can only be achieved up to a certain level. The important thing is that patients continued to observe a slight improvement, or it was maintained. Sometimes it happens that significant results regarding improvement are obtained after a year, so follow-ups were continued (also to intervene in case of unexpected disorders, admittedly, such disorders did not occur). Rehabilitation is a stimulus factor, and does not immediately act like pharmacotherapy. The most important thing is that the patients did not develop secondary disabilities, which is quite common after surgery. The sustained state of improvement testifies positively to both stem cell implantation and the effectiveness of physiotherapy.

Specific comments

 1. There is no data on  X ray  stage hip OA ? as well as on duration of the disease , previous treatments,  others localisations of OA  etc .. it is a little short .... 

When qualifying patients for CD34+ stem cell treatment, stage II coxarthrosis was assumed according to the Kelgren-Lawrence scale. We have added this information to the manuscript.

The duration of the disease was not assessed due to the difficulty in precise assessment. Often, patients report to the doctor some time after the first symptoms have appeared. Therefore, this parameter would not be reliable.

Previous treatment was only based on emergency pain relief.

The patients were diagnosed only with osteoarthritis of the hip joints, other degenerative diseases of the lower limb were not diagnosed.

2. How many patients with unilateral versus bilateral hip OA

60 patients with unilateral and 11 with bilateral coxarthrosis were qualified for the therapy.

3. What's mean rehabilitaion process ? it consists in which kind of t program in terms of time, type of exercices and frequency ?

Rehabilitation consisted of 3 parts. 1. Hospital rehabilitation (7 days) based on the use of a CPM (Continuous Passive Motion) splint, non-weight-bearing exercises, solux lamp (blue light). In total approx. 1.5 h per day. 2. Outpatient rehabilitation: (6 weeks) based on the use of exercises improving hip joint mobility, active exercises with resistance, quadriceps electrotherapy. In total approx. 1.5 h per day, 5 times per week. 3. Home rehabilitation (rest of the time) continuation of learned exercises. In total approx. half an hour per day, 5 times per week.

We added this information to the manuscript.

4. Was intake of others painkillers= , NSAIDS  recorded?

After CD34+ stem sell treatment, no NSAIDs were taken for 6 weeks (recommended after the procedure). Later, this parameter was not monitored, but paracetamol was recommended in case of pain.

5. What do we know excatly about the HSS sensitivity to change ?

According to the Cohen’s rule, an ES of 0.20–0.49 represents a small change, 0.50–0.79 a medium change, and ≥ 0.80 a large change. The SRM is the mean change in the patient score divided by the SD of the changed scores.

We find this information in the article: Singh JA, Schleck C, Harmsen S, Lewallen D. Clinically important improvement thresholds for Harris Hip Score and its ability to predict revision risk after primary total hip arthroplasty. BMC Musculoskelet Disord. 2016 Jun 10;17:256. doi: 10.1186/s12891-016-1106-8.

Reviewer 2 Report

Comments and Suggestions for Authors

the topic is relevant to the field as regenerative medicine represents one of the next goals of orthopaedics.   The limitation of the study is not having a comparison group.  Although the study is interesting, I recommend continuing with a more in-depth study and comparing more treatment groups. The topic is currently extremely debated so I recommend continuing with this kind of studies .

Author Response

Dear Reviewer,

thank you very much for your time on the review.

The lack of a control group has been included in the limitations section.

We are planning further research in this area.

Best regards,